# Sampling-Based Techniques for Training Deep Neural Networks with Limited Computational Resources: A Scalability Evaluation

**Sana Ebrahimi**                                                                 *sebrah7@uic.edu*
*Department of Computer Science*
*University of Illinois Chicago*

**Rishi Advani**                                                                  *radvani2@uic.edu*
*Department of Computer Science*
*University of Illinois Chicago*

**Abolfazl Asudeh**                                                               *asudeh@uic.edu*
*Department of Computer Science*
*University of Illinois Chicago*

## Abstract

Deep neural networks are superior to shallow networks in learning complex representations. As such, there is fast-growing interest in utilizing them in large-scale settings. The training process of neural networks is already known to be time-consuming, and having a deep architecture only aggravates the issue. This process consists mostly of matrix operations, among which matrix multiplication is the bottleneck. Several sampling-based techniques have been proposed for speeding up the training time of deep neural networks by approximating the matrix products. These techniques fall under two categories: (i) sampling a subset of nodes in every hidden layer as active at every iteration and (ii) sampling a subset of nodes from the previous layer to approximate the current layer's activations using the edges from the sampled nodes. In both cases, the matrix products are computed using only the selected samples. In this paper, we evaluate the scalability of these approaches on CPU machines with limited computational resources. Making a connection between the two research directions as special cases of approximating matrix multiplications in the context of neural networks, we provide a negative theoretical analysis that shows feedforward approximation is an obstacle against scalability. We conduct comprehensive experimental evaluations that demonstrate the most pressing challenges and limitations associated with the studied approaches. We observe that the hashing-based node selection method is not scalable to a large number of layers, confirming our theoretical analysis. Finally, we identify directions for future research.

## 1 Introduction

Deep neural networks (DNNs) have become a popular tool for a wide range of machine learning tasks (Liu et al., 2017), including image classification (Hemanth & Estrela, 2017), natural language processing (Otter et al., 2020; Lauriola et al., 2022; Kamath et al., 2019), and speech recognition (Kamath et al., 2019; Zhang et al., 2018). Recent advancements in DNNs have led to revolutionary solutions for traditionally challenging problems across different fields of science from health care (Miotto et al., 2018) and biology (Angermueller et al., 2016) to physics (Tanaka et al., 2021) and astronomy (Meher & Panda, 2021). The benefits of DNNs have reached many of the diverse areas of research in computer science, including data management (Kumar et al., 2017; Wang et al., 2016; Zhou et al., 2020), offering state-of-the-art approaches for a variety of tasks such as entity resolution (Li et al., 2020; 2021; Thirumuruganathan et al., 2021), cardinality estimation

(Hasan et al., 2020; Wu & Cong, 2021), and approximate query processing (Thirumuruganathan et al., 2020; Ma et al., 2021).

One of the crucial factors in developing models with high performance is the network architecture. The task and dataset at hand determine the most appropriate architecture to use. In addition to a large number of layers, DNNs often have a high number of nodes per layer. While large models can better generalize, training them can be computationally expensive, requiring extensive amounts of data and powerful hardware, including expensive GPUs. On the other hand, "the ubiquity of CPUs provides a workaround to the GPU's dominance" Smith (2023), motivating "democratiz[ing] AI with CPUs" (Shrivastava as quoted by Smith (2023)). Nevertheless, limited resources on personal computers with ordinary CPUs or mobile devices leads to difficulties in training DNNs to a sufficient level of accuracy. DNNs need to compute "activation values" for every layer in a forward pass and calculate gradients to update weights in the backpropagation. This requires performing computationally expensive matrix multiplications that make the training process inefficient. Furthermore, large matrices often do not fit in the cache, and storing them in main memory necessitates constant communication between the processor and memory, which is even more time consuming.

In this work, we explore the scalability of two directions in sampling-based approaches for efficient training of DNNs that can be applied on memory- and computation-constrained devices. Our contributions can be summarized as follows.

- We make a connection between two separate sampling-based research directions for training DNNs by showing that both techniques can be viewed as special cases of matrix approximation, where one samples rows of the weight matrix while the other sample its columns. To the best of our knowledge, there is no previous work in the literature to make this observation.

- After careful exploration of different techniques, we provide negative theoretical results that show estimation errors during the feedforward step propagate across layers. In particular, for the model of Spring & Shrivastava (2017), we prove that estimation error increases *exponentially* with the number of hidden layers.

- We provide extensive experiments using five training approaches and six benchmark datasets to evaluate the scalability of sampling-based approaches. Our experimental results confirm our theoretical analysis that feedforward approximation is an obstacle against scalability. In addition to other findings, our experiments reveal that while the model of Adelman et al. (2021) is scalable for mini-batch gradient decent when the batch size is relatively large, there is a research gap when it comes to designing scalable sampling-based approaches for stochastic gradient decent.

The rest of our paper is organized as follows. We first discuss some of the potential benefits of training DNNs on CPU machines in §2, followed by related work in §3. We define the problem formulation in §4 and provide a taxonomy of sampling-based approaches for efficient DNN training. In §5 and §6, we discuss two of these approaches in further detail. We present our theoretical analysis in §7 and discuss an extension to convolutional neural networks in §8. Experiment details and takeaways are discussed in §9, §10, and §11, and we offer concluding remarks in §12.

## 2  Potential Benefits of Training DNN on CPU Machines

The pursuit of advancing Deep Neural Network (DNN) training on CPU machines unveils a compelling avenue replete with practical advantages. Below, we briefly explain some of these potential benefits:

**Abundance of CPU Machines.**  CPU-equipped personal computing devices, including PCs and smartphones, enjoy widespread availability and accessibility among a vast segment of the population. Remarkably, the computational potential of these devices often remains underutilized. Leveraging such resources for DNN training introduces the opportunity to conduct this computational-intensive task at no additional cost for personal endeavors. Furthermore, while individual devices possess limited capacity, their collective potential can effectively address a multitude of moderate-sized artificial intelligence (AI) challenges. Recognizing

this collective capability, recent endeavors have emerged to design client-side AI frameworks, exemplified by JavaScript packages like Tensorflow.js, facilitating machine learning on the client-side. Advancements in DNN training on CPU machines directly benefit these platforms.

**Independence from Backend Servers.**    Personalized AI necessitates the training, or at the very least, fine-tuning of machine learning models with user-specific data. Opting for DNN training on CPU machines renders this process independent of backend GPU servers. Instead of transmitting data to the server, each personal device can locally fine-tune the models using its own data. This approach instantly confers several additional advantages:

1. **Privacy:** By refraining from transmitting data beyond the confines of personal devices, concerns regarding data privacy are substantially alleviated.

2. **Reduction in Backend Computation Costs:** The computational burden, which would otherwise entail extensive computations at the backend server (one for each user), is shifted to the client side at no supplementary expense to the server.

3. **Elimination of Network Dependency:** By localizing computations, the need for communication with a backend server becomes obsolete. This proves especially advantageous for users with limited internet access or in regions where network services are less reliable.

**Democratizing DNN Training.**    GPU-equipped machines, while gradually becoming more affordable, still pose a considerable financial barrier. These costs manifest in the form of GPU access or enterprise APIs, especially in the context of large models like ChatGPT. Consequently, such resources remain inaccessible to a significant portion of the population. The facilitation of DNN training on CPU machines effectively dismantles this accessibility barrier.

**Environmental Sustainability.**    Energy consumption during DNN training raises valid environmental concerns. Conventional services of this nature contribute to heightened energy usage. In contrast, efficient computation on CPUs, coupled with a reduction in data movement volume, holds the potential to mitigate energy consumption and thereby alleviate its adverse impact on global warming, provided that such practices gain global adoption.

Incorporating these considerations into the discourse of DNN training on CPU machines not only enriches the academic discussion but also underscores the profound implications of this research direction in addressing pressing real-world challenges and democratizing AI accessibility. It is evident that significant research efforts have been judiciously directed towards the principal trajectory of Deep Neural Network (DNN) training on GPU systems. Conversely, the avenue of training DNNs on CPU machines remains relatively underexplored within the research landscape.

## 3 Related Work

The increasing importance of DNN applications opened the door to a variety of challenges associated with training these models. While there are numerous works on techniques for scaling DNNs, many of them have expensive hardware requirements and use GPUs to accelerate training (Fatahalian et al., 2004). Unlike GPUs, CPUs are available on any device, so optimizing training performance on CPUs is beneficial. There have been studies in which distributed, concurrent, or parallel programming on CPUs has been used to accelerate training (Spring & Shrivastava, 2017; Dean et al., 2012; Kalamkar et al., 2020; Han et al., 2016; He et al., 2018; Vanhoucke et al., 2011), but these methods are not always generally applicable due to variation in hardware requirements. Hence, algorithms focused on algorithmic optimization of feedforward and backpropagation are essential. Usually, methods with little to no special hardware requirements are preferred. Several algorithms in the literature apply a variety of sampling-based (Spring & Shrivastava, 2017; Li et al., 2016; Ba & Frey, 2013; Adelman et al., 2021; Gale et al., 2019; Ma et al., 2019) or non-sampling-based approximations (Makhzani & Frey, 2015; Zhu et al., 2018; Marinò et al., 2023; Han et al., 2016) to improve training for DNNs.

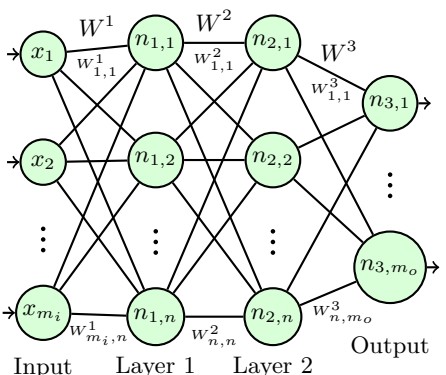

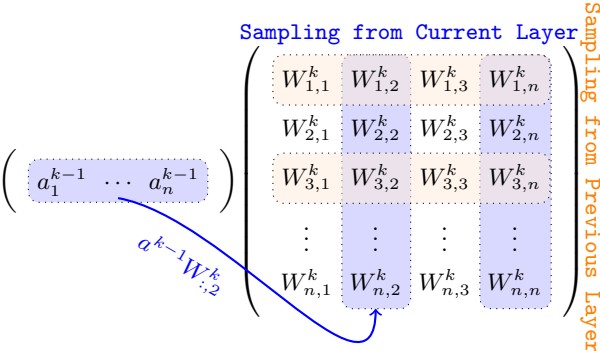

Figure 1: Neural network with $\ell = 3$ layers and $n$ nodes per hidden layer.

Figure 2: High-level idea of the sampling-based techniques.

Table 1: Table of Notations

| Notation | Description |
|----------|-------------|
| $\hat{y}$ | model output (prediction) |
| $y$ | target variable (ground truth) |
| $a^k$ | output vector of $k$-th layer |
| $z^k$ | input vector of the $k$-th layer |
| $W^k$ | weight matrix of the $k$-th layer |
| $W^k_{:,j}$ | column $j$ of $W^k$ |
| $W^k_{i,:}$ | row $i$ of $W^k$ |
| $\mathcal{L}(\hat{y}, y)$ | loss function |
| $\langle x_1, x_2 \rangle$ | inner product of $x_1$ and $x_2$ |
| $\|x\|$ | $\ell_2$ norm of vector $x$ |
| $\|X\|_F$ | Frobenius norm of matrix $X$ |

Several studies have shown that one way to scale up DNNs is to simplify the matrix-matrix or vector-matrix operations involved (Srivastava et al., 2014; Ba & Frey, 2013; Wang & Manning, 2013; Adelman et al., 2021; Spring & Shrivastava, 2017). The complexity of matrix multiplication dominates the computational complexity of training a neural network. The multiplication of large matrices is known to be the main bottleneck in training DNNs. Often, we try to sparsify the matrices, which can minimize the communication between the memory and the processors (Yao et al., 2023). Pruning the network and limiting the calculations in both directions to a subset of nodes per layer is one solution to train DNNs efficiently. This is what dropout-type algorithms suggest (Srivastava et al., 2014; Ba & Frey, 2013; Gale et al., 2019). Dropout-type methods either use a data-dependent sampling distribution (Ba & Frey, 2013; Wang & Manning, 2013; Adelman et al., 2021) or a predetermined sampling probability (Srivastava et al., 2014; Spring & Shrivastava, 2017). Note that these techniques are able to provide a good approximation only if used in the context of neural networks; they are not necessarily applicable to general matrix multiplication.

## 4 Preliminaries

In this section, we describe the neural network model, feedforward step, and backpropagation in the form of matrix operations. Finally, we discuss state-of-the-art sampling-based algorithms.

### 4.1 Problem Description

Many neural network architectures have been studied over the past decade. In this paper we focus on the standard multi-layer perceptron (MLP) model and analyze two major directions of sampling-based techniques for training neural networks.

Consider a feedforward neural network with $m_i$ inputs, $m_o$ outputs, and $\ell$ hidden layers. In general, every hidden layer $k$ contains $n_k$ hidden nodes while the nodes in the $(k-1)$-th and $k$-th layers are fully connected. Without loss of generality, for ease of explanation, we assume all hidden layers have exactly $n$ hidden nodes (Figure 1). For each layer $k$, we denote the vector of outputs by $a^k \in \mathbb{R}^{1 \times n}$. Similarly, $W^k \in \mathbb{R}^{n \times n}$ and $b^k \in \mathbb{R}^{1 \times n}$ are the weights and the biases of layer $k$,[1] respectively.

---

[1]The first and last layer are exceptions: $W^1 \in \mathbb{R}^{m_i \times n}$ and $W^{\ell-1} \in \mathbb{R}^{n \times m_o}$.

Let $f$ be the activation function (e.g., sigmoid or ReLU). With input vector $x \in \mathbb{R}^{1 \times m_i}$, the feedforward step is a chain of matrix products and activation functions that maps input vector $x$ to output vector $y$ and can be represented using the following equations:

$$a^0 = x \qquad z^k = a^{k-1}W^k + b^k \quad \forall k \in \{1, \ldots, \ell\} \qquad a^k = f(z^k) \tag{1}$$

In the setting described above, matrix-vector multiplication can be done in $\Theta(n^2)$ time and applying the element-wise activation function takes $\Theta(n)$ time for each layer. Thus, the entire feedforward process for the whole network is in the order of $\Theta(\ell n^2)$.

The final aspect in the training of neural networks is backpropagation, an efficient method of computing gradients for the weights to move the network towards an optimal solution via stochastic gradient descent[2] (SGD) (Goodfellow et al., 2016). Following Equation 1, the weight gradients for the backpropagation step can be computed recursively using Equation 2, where $\mathcal{L}$ is the loss function and $\odot$ is the Hadamard product.

$$
\begin{aligned}
\delta^\ell &= \nabla_{z^\ell}\mathcal{L} = f'(z^\ell) \odot \nabla_{a^\ell}\mathcal{L} & \nabla_{W^k}\mathcal{L} &= a^k\delta^{k+1} \\
\delta^k &= \nabla_{z^k}\mathcal{L} = f'(z^k) \odot W\delta^{k+1} & \nabla_{b^k}\mathcal{L} &= \delta^{k+1}
\end{aligned}
\tag{2}
$$

With gradient $a^k\delta_{k+1}$ and learning rate $\eta$, the weight matrix $W^k$ will be updated to $W^k - \eta a^k\delta_{k+1}$. The gradient computation and update operations are also in form of vector-matrix operations that take $\Theta(n^2)$ time for each layer. As a result, the backpropagation step in SGD also requires $\Theta(\ell n^2)$ time.

## 4.2 Taxonomy of Sampling-Based Techniques

The computation bottleneck in the training of a DNN is matrix multiplication, in form of a $(1 \times n)$ to $(n \times n)$ vector-matrix product for SGD. Sampling-based approaches seek to speed up this operation by skipping a large portion of the scalar operations. SGD is a noisy algorithm by nature. As such, it is more tolerant of small amounts of noise (Markidis et al., 2018), allowing for approximation. At a high level, these approaches fall in two categories, as shown in Figure 3.

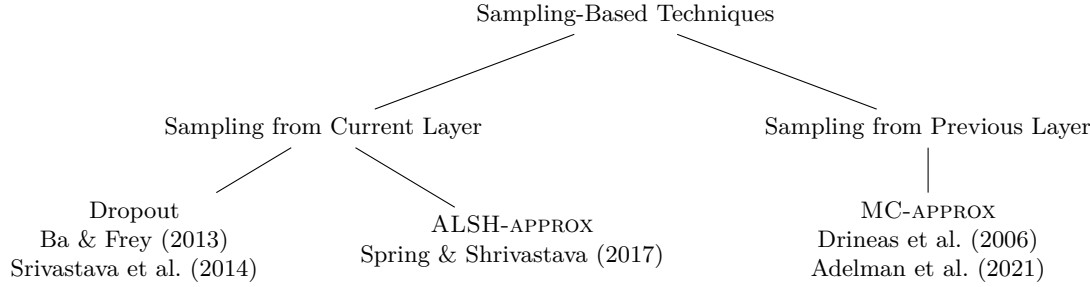

Figure 3: A taxonomy of sampling-based techniques for training DNNs

**Sampling from Current Layer.** The approaches in this category select a small subset of nodes in each layer during each feedforward–backpropagation step, and update the values of only those nodes. From Figure 1, recall that each column $W_{:,j}^k$ corresponds to the node $n_j^k$ in layer $k$, while each cell $W_{i,j}^k$ in that column represent the weight of the edge from $n_i^{k-1}$ to $n_j^k$. As a result, these approaches can be viewed as *selecting a small subset of the columns* of $W^k$ (e.g., highlighted columns in Figure 2) and conducting the inner product only for those.

---

[2]While SGD uses only one data point to compute the gradients, an alternative approach is mini-batch gradient descent (MGD), where a small sample set (mini-batch) of the training set is used for estimations. Note that SGD can be viewed as a special case of MGD where the batch size is 1. Following our scope in this paper, SGD is considered for problem formulation, explaining the learning algorithm, and analysis. Nevertheless, as we shall later explain in §6.1, one of the evaluated approaches, MC-APPROX, is based on MGD, the generalization of SGD. While SGD operations are in form of vector to matrix multiplication, MGD operations are in form of matrix (vectors of samples in the mini-batch) to matrix multiplication.

**Sampling from Previous Layer.** Instead of selecting a subset of columns and computing the exact inner-product for them, the alternative is to select all columns but compute the inner-product approximately for them by *selecting a small subset of rows* of $W^k$ (e.g., highlighted rows in Figure 2). That is, instead of computing the sum for all $n$ scalars in an inner-product, to *estimate* the sum by sampling a small number of scalars.

Next, in §5 and §6 we provide a detailed exploration of the representative approaches in each category.

## 5 Efficient Training by Sampling from Current Layer

### 5.1 Dropout

Srivastava et al. (2014) introduced DROPOUT, a computationally efficient training approach that reduces the risk of overfitting. During each feedforward step, the algorithm picks a subset of nodes uniformly at random in each hidden layer and drops the remaining nodes temporarily. The sampled nodes are then used for feedforward evaluation and backpropagation.

While DROPOUT was originally introduced to fix overfitting, it introduced a computation reduction to the training process. In many cases, DROPOUT improved the runtime efficiency compared to the standard training process on the same architecture. However, there are scenarios in which training under DROPOUT requires more training iterations and eventually hurts the runtime. One can observe that due to the randomness in sampling with a fixed probability (usually $p = 1/2$), there is a risk of dropping nodes that significantly affect the output values. Ba & Frey (2013) addressed this issue by proposing ADAPTIVE-DROPOUT, which uses a data-dependent distribution that is an approximation of the Bayesian posterior distribution over the model architecture and updates the sampling ratio adaptively w.r.t the current network. This method avoids randomly dropping significant nodes in the model.

### 5.2 Asymmetric Locality-Sensitive Hashing for Training Approximation

Unlike in DROPOUT, one might want to intelligently select a small subset of so-called *active nodes* for each layer for computing the inner products. In particular, given the vector $a^{k-1}$, the goal is to find a small portion of nodes $j$ in layer $k$ for which the value of $a^{k-1}W_{:,j}^k$ is maximized in order to avoid computing inner products for small values (estimating them as zero). Given a set $S$ of vectors (in this case, the set of columns in $W^k$) and a query vector $a$, the problem of finding a vector $w^* \in S$ with maximum inner product $\langle a, w^* \rangle$ is called maximum inner-product search (MIPS). To solve MIPS, Shrivastava & Li (2014) employ asymmetric locality-sensitive hashing (ALSH).

**Definition 1** (Asymmetric Locality-Sensitive Hashing (Shrivastava & Li, 2014))**.** Given a similarity threshold $S_0$ and similarity function $\text{sim}(\cdot)$, a family $\mathcal{H}$ of hash functions are $(S_0, cS_0, p_1, p_2)$-sensitive for $c$-NNS[3] with $a \in \mathbb{R}^n$ as query and a set of $w \in \mathbb{R}^n$ vectors if for all $h \in \mathcal{H}$ chosen uniformly, the following conditions are satisfied:

$$
\begin{aligned}
\text{sim}(w, a) \geq\ & S_0 &&\implies \Pr[h(Q(a)) = h(P(w))] \geq p_1 \\
\text{sim}(w, a) \leq\ & cS_0 &&\implies \Pr[h(Q(a)) = h(P(w))] \leq p_2
\end{aligned}
$$

For $w, a \in \mathbb{R}^n$ with $\|w\| \leq C$, where $C$ is a constant less than 1, and $\|a\| = 1$, they define the transformations $P$ and $Q$ for the inner product as follows.

$$
\begin{aligned}
P\colon \mathbb{R}^n \to \mathbb{R}^{n+m}, \quad & w \mapsto \left[w; \|w\|^{2^1}, \ldots, \|w\|^{2^m}\right] \\
Q\colon \mathbb{R}^n \to \mathbb{R}^{n+m}, \quad & a \mapsto \left[a; 1/2, \ldots, 1/2\right]
\end{aligned}
\tag{3}
$$

In other words, to generate $P$, $w$ is padded with $m$ terms, where term $i$ is the $\ell_2$ norm of $w$ to the power of $2^i$. $Q$ is generated by padding $a$ with $m$ copies of the constant $1/2$. Shrivastava & Li (2014) prove that NNS

---

[3]$c$-approximation of nearest neighbor search Indyk & Motwani (1998)

in the transformed space is equivalent to the maximum inner product in the original space:

$$\arg\max_{w}\langle w, a\rangle \approx \arg\min_{w}\|Q(a) - P(w)\|. \tag{4}$$

Equation 4 motivates using MIPS for efficient training of DNNs. Spring & Shrivastava (2017) build their algorithm (referred to as **ALSH-approx** in this paper) upon Equation 4. As explained in §4, the feedforward step and backpropagation consist of many matrix multiplications, each of which involve a set of inner products as large as each hidden layer. ALSH-APPROX uses ALSH to prune each layer by finding active nodes, in this case, nodes with maximum activation values. This is equivalent to solving MIPS in each layer.

Essentially, ALSH-APPROX uses ALSH to find active nodes $j$ whose weight vector $W_{:,j}^k$ collides with an input vector $a^{k-1}$ under the same hash function. The probability of collision captures the similarity of vectors in each hidden layer. To do so, it sets the query vector as $q = a^{k-1}$ and the set of vectors using the columns of $W^k$ as $S = \left\{ W_{:,1}^k, \ldots, W_{:,n}^k \right\}$. Then, after constructing $Q$ and $P$ based on Equation 3, we have

$$\arg\max_{j}\langle W_{:,j}^k, a^{k-1}\rangle \approx \arg\min_{j}\|Q(a^{k-1}) - P(W_{:,j}^k)\|. \tag{5}$$

ALSH-APPROX constructs $L$ independent hash tables with $2^K$ hash buckets and assigns a $K$-bit randomized hash function to every table. Each layer has been assigned $L$ hash tables and a meta hash function to compute a hash signature for the weight vectors and fill all the hash tables before training. In this setting, $K$ and $L$ are tunable hyperparameters that affect the active set's size and quality.

During training, ALSH-APPROX computes the hash signature of each incoming input using the existing hash functions. Then, a set of weight vectors will be returned using the hash values corresponding to the hash bucket. The active nodes in a layer are the union of their corresponding weight vectors from probing $L$ hash tables. Then, the model only performs the exact inner product on these active nodes and skips the rest. Finally, the gradient will only backpropagate through the active nodes and update the corresponding weights. In other words, ALSH is used to sample a subset of nodes with probability $1 - (1 - p^K)^L$ if $p$ is the probability of collision.

Updating the hash tables ensures that the modified weight vectors are recognized. Based on the results reported by Spring & Shrivastava (2017), the number of active nodes for each input can be as small as 5% of the total nodes per layer. Thus, ALSH-APPROX performs a significantly smaller set of inner products in each iteration. Moreover, due to the sparsity of the active sets belonging to different data inputs, the overlap between them throughout the dataset is small. Accordingly, the weight gradient updates corresponding to these inputs are sparse as well. To leverage this, the hash table updates are executed after processing a batch of inputs and can be executed in parallel. The main advantage of ALSH-APPROX is that, unlike DROPOUT, it finds the active nodes *before* computing the inner products.

## 6 Efficient Training by Sampling from Previous Layer

While techniques discussed in §5 reduce the vector-matrix multiplication time by selecting a subset of columns (nodes) from each weight matrix $W^k$ and computing the inner product exactly for them, an alternative approach is to select all columns but to compute inner products approximately. This idea has been proposed by Adelman et al. (2021). This paper is built on the Monte Carlo (MC) method by Drineas et al. (2006) for fast approximation of matrix multiplication. We first review the work of Drineas et al. (2006) in §6.1 and then in §6.2 we explain how Adelman et al. (2021) adapt the method to develop an algorithm for efficient training of DNNs.

### 6.1 Fast Approximation of Matrix Multiplication

For many applications, a fast estimation of the matrix product is good enough. In addition to hardware/software oriented optimizations such as cache management (Fatahalian et al., 2004; Goto & Geijn, 2008) or half precision computations (Markidis et al., 2018; Vanhoucke et al., 2011), Monte Carlo (MC) methods

(Robert, 2016) have been proposed for such estimations. At a high level, MC methods use repeated sampling and the law of large numbers to estimate aggregate values.

Recall that given two matrices $A \in \mathbb{R}^{m \times n}$ and $B \in \mathbb{R}^{n \times p}$, the product $AB$ is an $m \times p$ matrix, where every element $AB_{i,j}$ is the inner product of $i$-th row of $A$ with the $j$-th column of $B$:

$$AB_{i,j} = \langle A_{i,:}^T, B_{:,j} \rangle = \sum_{t=1}^{n} A_{i,t} B_{t,j} \tag{6}$$

In an MC estimation of $AB_{i,j}$, instead of computing the sum over all $t \in [1, n]$, only a small sample of elements $\sigma \subset \{1, \ldots, n\}$, where $c = |\sigma| \ll n$, are considered. Arguing that uniform sampling would add a high error in estimating $AB$, Drineas et al. (2006) introduce a nonuniform sampling method to generate $\sigma$ with a probability proportional to the magnitude of data. Specifically, they develop a randomized algorithm that samples each column $i$ of A and row $i$ of B with probability

$$p_i = \frac{\|A_{:,i}\| \cdot \|B_{i,:}\|}{\sum_{t=1}^{n} \|A_{:,t}\| \cdot \|B_{t,:}\|}. \tag{7}$$

They define $C \in \mathbb{R}^{m \times c}$ and $R \in \mathbb{R}^{c \times p}$ by $C = ASD$ and $R = (SD)^T B$, respectively, where $S$ is an $n \times c$ sampling matrix where $S_{ij} = 1$ if the $i$-th row of $A$ is the $j$-th sample and $D$ is a $c \times c$ diagonal scaling matrix in which $D_{jj} = \frac{1}{\sqrt{cp_j}}$. The authors prove that defining $p_i$ as in Equation 7 minimizes the expected estimation error $\mathbb{E}\big[\|AB - CR\|_F\big]$. Then each element $AB_{i,j}$ is estimated as $\sum_{t=1}^{c} C_{i,t} R_{t,j} = \sum_{t=1}^{c} \frac{1}{cp_i} A_{i,t} B_{t,j} \approx AB_{i,j}$. Sampling row-column pairs w.r.t $p_i$ reduces the time complexity of matrix multiplication to $O(mcp)$.

## 6.2 MC-approx

Training DNNs involves computationally expensive matrix multiplication operations. However, the gradients computed during backpropagation only approximate directions towards the optimal solution, so the training process has a high tolerance to small amounts of noise. This makes approximation of matrix multiplication a reasonable choice to speed up training of DNNs. Following this idea, Adelman et al. (2021) propose a MC approximation method for fast training of DNNs (in this paper, referred to as **MC-approx$_\mathbf{M}$** for the mini-batch setting and **MC-approx$_\mathbf{S}$** for the stochastic setting) based on the MC estimation of matrix multiplication explained in §6.1. Despite the fact that Drineas et al. (2006) provide an unbiased estimate for the matrix multiplication $AB$ (i.e., $E[CR] = AB$), Adelman et al. (2021) prove that the sampling distribution is not able to provide an unbiased estimation of the weight gradient if it is used for both the forward step and backward pass simultaneously.

One way to eliminate the bias is to use MC approximation only in forward pass, propagate the gradient through the entire network, and perform the exact computations. However, experiments show this approach results in lower accuracy in practice. Therefore, Adelman et al. (2021) propose a new sampling distribution that yields an unbiased estimate of the weight gradient $\nabla_W \hat{\mathcal{L}}$ when it is used only during the feedforward step. Specifically, they sample column-row pairs independently from $A \in \mathbb{R}^{m \times n}$ and $B \in \mathbb{R}^{n \times p}$.

Let $k$ be the number of samples for estimation, let $V \in \mathbb{R}^{n \times n}$ be a diagonal sampling matrix with $V_{i,i} = Z_i \sim \text{Bernoulli}(p_i)$, where $\sum_{i=0}^{n} p_i = k$, and let $D \in \mathbb{R}^{n \times n}$ be a diagonal scaling matrix with $D_{i,i} = \frac{1}{\sqrt{p_i}}$. Then, $AB \approx \sum_{i=0}^{n} \frac{Z_i}{p_i} A_{:,i} B_{i,:} = AVDD^T V^T B = A'B'$, and the estimation error is $E\big[\|AB - A'B'\|_F^2\big] = \sum_{i=0}^{n} \frac{1-p_i}{p_i} \|A_i\|^2 \|B_i\|^2$. Under the constraint $\sum_{i=0}^{n} p_i = k$, the estimation error is minimized by

$$p_i = \min\left\{ \frac{k\|A_i\|\|B_i\|}{\sum_{t=0}^{n} \|A_t\|\|B_t\|}, 1 \right\} \tag{8}$$

The authors prove that training a neural network by approximating matrix products in backpropagation converges with the same rate as standard SGD and results in an unbiased estimator when nonlinearities are not considered. When accounting for nonlinearities, the results hold as long as the MC approximation of $Wx$ is unbiased and the activation and loss functions are $\beta$-Lipschitz.

## 7 Theoretical Analysis

As we explained in §5.2 and §6.1, sampling-based approaches seek to speed up the training of DNNs by skipping a large number of computations and only approximate matrix multiplications. In this section we provide negative theoretical results for scalability against the feedforward approximation. At a high level, we show that small estimation errors in the initial layers get propagated and compounded in subsequent layers. Adelman et al. (2021) already observed the low performance of MC-APPROX when the feedforward step is approximated and therefore only applied approximation during backpropagation for MLPs. As such, we focus on ALSH-APPROX for our analysis. First, let us introduce the following notation, which we will use throughout this section.

- $\bar{a}_j^k$: the estimation of $a_j^k$ by ALSH-APPROX.

- $e_j^k = a_j^k - \bar{a}_j^k$: the activation value estimation error.

- $\uparrow_j^k$: the set of active nodes for $n_j^k$.

**Lemma 1.** *Let $f$ be a linear activation function such that $a = f(z) = z$. Assuming the active nodes are detected exactly, the estimation error for the node $n_j^k$ by ALSH-APPROX is as follows.*[4]

$$e_j^k = \begin{cases} \sum\limits_{i \notin \uparrow_j^1} x_i W_{i,j}^1 & \text{if } k = 1 \\ e^{k-1} W_{:,j}^k + \sum\limits_{i \notin \uparrow_j^k} \bar{a}_i^{k-1} W_{i,j}^1 & \text{otherwise} \end{cases}$$

Lemma 1 provides a recursive formula for the activation value estimation error in terms of the weighted summation over active nodes versus inactive nodes. To provide a non-recursive and easier to understand formula, in Theorem 2 we assume that the weighted summation over the active nodes is always $c$ times that of the inactive nodes.

**Theorem 2.** *Let $f$ be a linear activation function such that $a = f(z) = z$. Suppose for any node $n_p^l$,*

$$\sum_{i \in \uparrow_p^l} a_i^{l-1} W_{i,p} = c \sum_{i \notin \uparrow_p^l} a_i^{l-1} W_{i,p} \,.$$

*Then, $a_j^k = \bar{a}_j^k \left(\frac{c+1}{c}\right)^k$. Therefore, $e_j^k = \bar{a}_j^k \left(\left(\frac{c+1}{c}\right)^k - 1\right)$.*

Theorem 2 proves that the estimation error increases **exponentially** with the number of layers. As a result, due to the sharp increase in the estimation error, ALSH-APPROX does not scale to DNNs. To better observe this, suppose $c = 5$ (i.e., the weighted sum for the active nodes is five times that of the inactive nodes). Then, using Theorem 2, the error-to-estimate ratios for different numbers of layers are as follows.

| **k** | 1 | 2 | 3 | 4 | 5 | 6 |
|---|---|---|---|---|---|---|
| $\mathbf{e_j^k/\bar{a}_j^k}$ | 0.2 | 0.44 | 0.72 | 1.07 | 1.48 | 1.98 |

From the above table, it is evident that as soon as the depth of the network gets larger than 3, the estimation error dominates the estimation value. This is consistent with our experiment results, where ALSH-APPROX failed to scale for networks with more than 3 hidden layers.

## 8 Extension to Convolutional Neural Networks

The focus of this paper, and the existing work, has mainly been on the fully connected networks (Figure 1), where every node in layer $k$ is connected to every node in layer $k-1$. Another popular architecture is the

---

[4]Proofs are provided in the appendix.

convolutional neural networks (CNN), which is often used for for grid-like data, such as image data. The key difference of CNNs is their core building block, the *convolutional layer*, which is *not* fully connected. Consider an image to be a $\mu \times \mu$ grid of pixels, passed to a CNN as the input vector with $n = \mu \times \mu$ values, and a kernel of size $n' = \nu \times \nu$, where $\nu \ll \mu$. The kernel can be viewed as a $\nu$ by $\nu$ patch swept over the image. As a result, each node in a convolutional layer is connected to only $n'$ nodes (pixels that fall inside the patch). It is equivalent of masking out $(n - n')$ values in the weight vector associated with each node in the convolution layer. Subsequently, the weight vector associated with a node will contain only a very small fraction, $\frac{\nu \times \nu}{\mu \times \mu}$, of unmasked values.

Now, let us recall from §4.2 and Figure 2 that the sampling-based techniques either (ALSH-APPROX) select a subset of column and compute the inner-product for them exactly or (MC-APPROX) select all columns but compute the inner-products approximately (by selecting a subset of rows). As a result, MC-APPROX can be extended to convolutional layers by using only the $n'$ edges (rows) from the previous layer to estimate the inner product for each node in the current layer (every column). In particular, MC-APPROX incorporates approximation exclusively within the convolutional layers, while maintaining exact computations in the classifier layers.

Conversely, extending ALSH-APPROX to CNNs is challenging. Recall that ALSH-APPROX uses locality-sensitive hashing to identify the active nodes: the nodes with weight vectors that have maximum inner-product with the input vector. In order to identify the active nodes, ALSH-APPROX *selects rows in $W^k$ from LSH buckets* for a layer $k$ based on their similarity with the input, which is detected by the collision of their hash values. On the other hand, for each node in a convolutional layer, only $n'$ of the rows are unmasked. As a concrete example, in our experiment on the CIFAR-10 dataset, each image is $32 \times 32$ pixels and the kernel is $3 \times 3$. In this setting, each node in a convolutional layer is connected only to 9 (0.8%) of the 1024 input pixels. The low percentage of unmasked rows for each node in a convolutional layer makes it unlikely that those nodes are among the random selections based on which the hash buckets are constructed.

For a node in a convolutional layer to be identified as active, it should fall into at least one of the $L$ LSH buckets that the input vector falls into. The LSH buckets are constructed using random sign projection as the hash function. That is, each hash function is a random hyperplane that partitions the search space in two halves (positive and negative sides). A set of $K$ such hyperplanes partitions the space into $2^K$ convex regions, each representing a bucket. As a result, unless the input vector has near-zero values on (a large portion of) the masked rows, it is unlikely that the corresponding node falls into the same bucket as the input vector. Such input vectors (images) are unlikely in practice.

In summary, the hash function that has been built for the entire set of $n$ rows is not an effective near-neighbor index for detecting active nodes for convolutional layers. Alternatively, one can create a separate LSH family, based on the unmasked weights, for each of the $n$ nodes in a convolutional layer; this approach, however, is not practical due to the time and memory overhead for building the hash functions.

Therefore, in order to extend ALSH-APPROX to CNNs, we only consider using LSH for the fully connected (classifier) layers, while maintaining exact computations in convolutional layers. Still, we empirically observed a decline in the model performance in our experiments. Specifically, Table 3 confirms that for a model with 4 convolutional layers with residual blocks (ResNet18) and 2 fully connected layers on the CIFAR-10 dataset, the accuracy dropped to 10.3% (almost as bad as random guessing).

## 9    Experiment Setup

### 9.1    Hardware

This paper aims to evaluate sampling-based approaches for training DNNs on regularly available machines; thus, we ran all experiments on a single-CPU machine (Intel Core i9-9920X machine with 128 GB of memory) without a GPU.

### 9.2 Datasets

We used six benchmark datasets for our experiments.

**MNIST (Deng, 2012)** 70,000 handwritten digits, each in the form of a $28 \times 28$ grayscale image, and 10 classes (digits zero to nine).

**Fashion-MNIST (Xiao et al., 2017)** 70,000 fashion products, each in the form of a $28 \times 28$ grayscale image, and 10 classes.

**EMNIST-Letters (Cohen et al., 2017)** 145,600 handwritten letters, each in the form of a $28 \times 28$ grayscale image, and 26 classes.

**Kuzushiji-MNIST (Clanuwat et al., 2018)** 70,000 cursive Japanese characters, each in the form of a $28 \times 28$ grayscale image, and 10 classes.

**NORB (Fu Jie Huang, 2005)** 48,600 photographs of 50 toys from different angles, each in the form of a $96 \times 96$ grayscale image, and 5 classes.

**CIFAR-10 (Krizhevsky, 2009)** 60,000 color images, each of dimensions $32 \times 32$, and 10 classes.

We randomly split the datasets into train, validation, and test sets as shown in Table 2. We obtained similar results across different datasets. For brevity, we provide a detailed discussion only using the results on the MNIST dataset, but extensive results for other datasets are provided in the appendix.

### 9.3 Methods Evaluated

We evaluated four sampling-based approaches for training DNNs discussed in §5 and §6, namely DROPOUT (Srivastava et al., 2014), ADAPTIVE-DROPOUT (Ba & Frey, 2013), MC-APPROX (Adelman et al., 2021), and ALSH-APPROX (Spring & Shrivastava, 2017), on fully connected DNNs. In addition, the regular training approach, referred to as STANDARD, has been implemented for comparison purposes. All implementations are in Python 3.9 using the PyTorch library. For MC-APPROX,[5] ALSH-APPROX,[6,7] DROPOUT,[8] and ADAPTIVE-DROPOUT[9] we used the publicly available code.

Table 2: Dataset splits

| Dataset | Train | Test | Validation |
|---|---|---|---|
| MNIST | 55000 | 10000 | 5000 |
| Fashion-MNIST | 55000 | 10000 | 5000 |
| Kuzushiji-MNIST | 55000 | 10000 | 5000 |
| EMNIST-Letters | 104800 | 20800 | 20000 |
| NORB | 22300 | 24300 | 2000 |
| CIFAR-10 | 45000 | 10000 | 5000 |

### 9.4 Default Values

To train our models, we use SGD. The activation function used for hidden layers is ReLU due to its simplicity, ease of computation, and the fact that it helps with the vanishing gradients problem (Goodfellow et al., 2016). The output layer activation function is log softmax, and the loss function used throughout experiments is the negative log-likelihood. The learning rate is always either $10^{-4}$ or $10^{-3}$ depending on the setting, and the models are trained for 50 epochs. In particular, we set the learning rate to $10^{-4}$ for MC-APPROX$_S$. The hyperparameters of our implementation are the best values reported for each approach. For MC-APPROX the batch size is set to 20 and $k = 10$. For ALSH-APPROX, $K = 6$, $L = 5$, and $m = 3$ (Equation 3) as specified in (Spring & Shrivastava, 2017). In order to have a fair comparison with ALSH-APPROX, we set the probability of picking nodes for DROPOUT and ADAPTIVE-DROPOUT to $p = 0.05$, and we use a network with 3 hidden layers and 1000 hidden units per layer across algorithms. The implementation of ALSH-APPROX

---

[5]`github.com/acsl-technion/approx`
[6]`github.com/rdspring1/LSH-Mutual-Information`
[7]`github.com/rdspring1/LSH_DeepLearning`
[8]`github.com/gngdb/adaptive-standout`
[9]see footnote 8.

provided in (Spring & Shrivastava, 2020) performs better when using the Adam optimizer (Kingma & Ba, 2014) than when using Adagrad (Duchi et al., 2011), which is used in the original implementation in (Spring & Shrivastava, 2017). Hence, we use Adam in our experiments.

For the convolutional setting, we used ResNet-18 with two fully-connected layers as a classifier to run our experiments. We limit the approximation to the classifier and keep the convoluted operations exact. Also, for CIFAR-10, we use pure SGD instead of Adam.

### 9.5   Evaluation Metrics

We use accuracy and time as evaluation metrics. Accuracy here refers to the percentage of correct predictions on the entire dataset. Since the task we focus on is multi-class classification, we also provide confusion matrices.

### 9.6   Experiment Plan

We are mainly interested in evaluating the following.

**Accuracy**  How do the algorithms perform when training networks with different depths?

**Time**  How scalable are the evaluated algorithms (in particular, ALSH-approx and MC-approx) w.r.t training time?

**Hyperparameters**  How do hyper-parameter choices (e.g., batch size) affect training time and accuracy?

## 10   Experiment Results

### 10.1   Scalability Evaluation: Accuracy

We begin our experiments by addressing the first question in §9.6. To do so, we generate models with different numbers of hidden layers (1 to 7) and evaluate each method on all six datasets discussed in §9.2 for both stochastic and mini-batch settings. The confusion matrices for all algorithms are provided in Figure 4. Every row in the figure shows the performance of an algorithm, while different columns represent networks with different numbers of hidden layers. In all plots contained within the figure, the x-axis shows the model prediction and the y-axis shows the true labels. Consequently, the diagonal cells show correct predictions, while all other cells are incorrect predictions. Ideally, the models should have (near-)zero values on non-diagonal cells.

Table 3: Test accuracy (%) for a network with 3 hidden layers.[10]

| Dataset | ALSH-approx | MC-approx$_M$ | MC-approx$_S$ | Dropout$_S$ | Adaptive-Dropout$_S$ | Standard$_S$ |
|---|---|---|---|---|---|---|
| MNIST | 94.15 | 98.10 | **98.38** | 90.21 | 98.06 | 96.46 |
| Kuzushiji-MNIST | 72.87 | 91.78 | **96.50** | 9.84 | 90.73 | 83.86 |
| Fashion-MNIST | 78.11 | 87.85 | **88.58** | 76.28 | 86.12 | 73.64 |
| EMNIST-Letters | 64.97 | 89.84 | **90.75** | 4.96 | 89.50 | 85.34 |
| NORB | 78.57 | 92.05 | **97.52** | 32.73 | 96.60 | 51.61 |
| CIFAR-10 | 10.31 | 73.26 | 62.11 | 67.85 | 75.55 | **93.02** |

**Baselines.**  Standard, Dropout, and Adaptive-Dropout (the first three rows in Figure 4) are our baselines for comparisons. For Dropout, the nodes are sampled randomly with probability $p$, and for Adaptive-Dropout, $p$ is updated w.r.t the Bayesian posterior distribution of data input. In standard feedforward training, we expect to observe a decrease in generalization error over complex datasets as we

---

[10]Across all tables and plots we use the subscripts "S" and "M" to refer to the SGD and mini-batch SGD (with default batch size 20) settings, respectively. When there is no subscript, the default is MGD for MC-approx and SGD for all other methods.

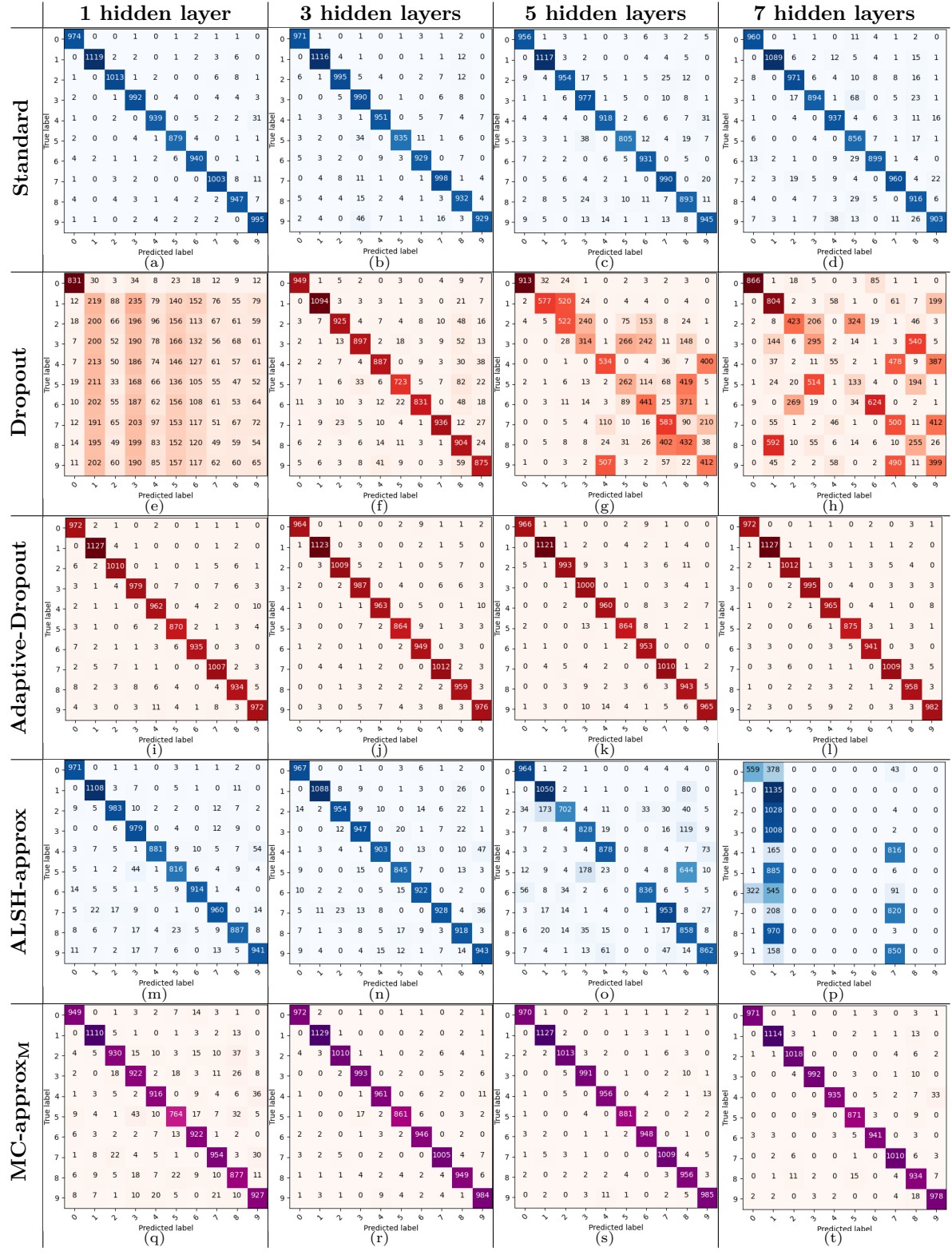

Figure 4: Confusion matrices of different algorithms for different numbers of hidden layers. In all plots, x-axis and y-axis are the predicted and true labels (0 to 9), respectively.

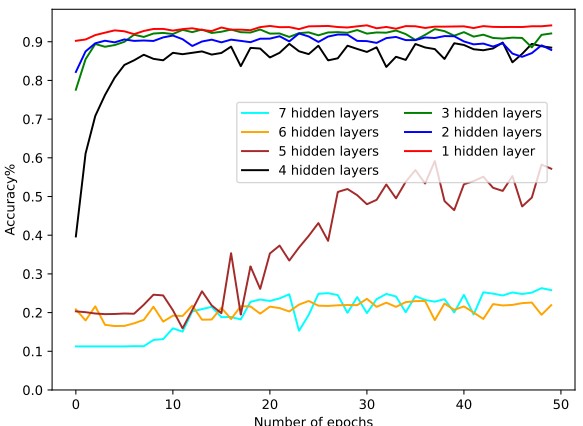

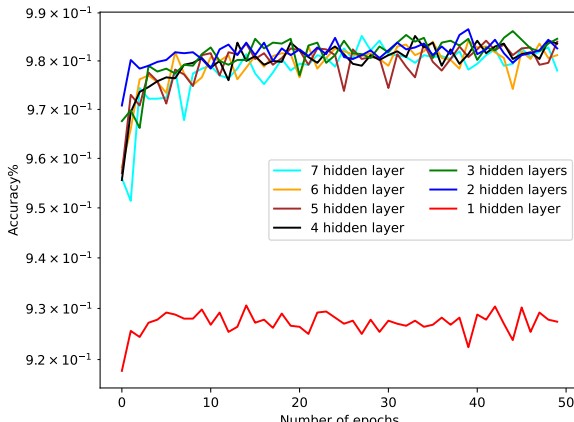

Figure 5: ALSH-APPROX: validation accuracy for different numbers of layers.

Figure 6: MC-APPROX: validation accuracy for different numbers of layers.

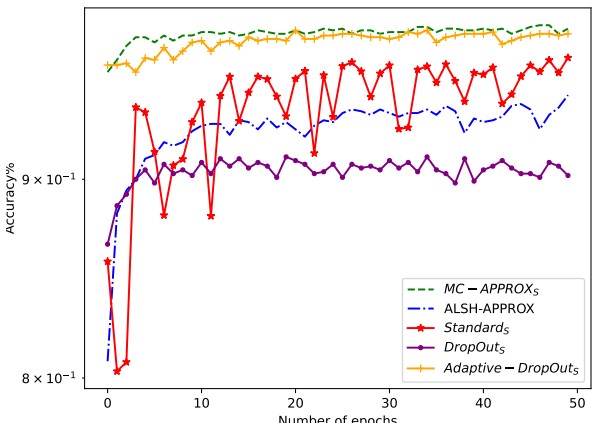

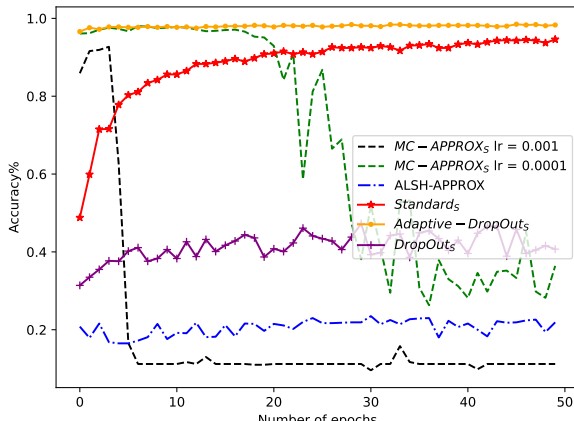

Figure 7: Validation accuracy for each method with 3 hidden layers.

Figure 8: Validation accuracy for each method with 7 hidden layers.

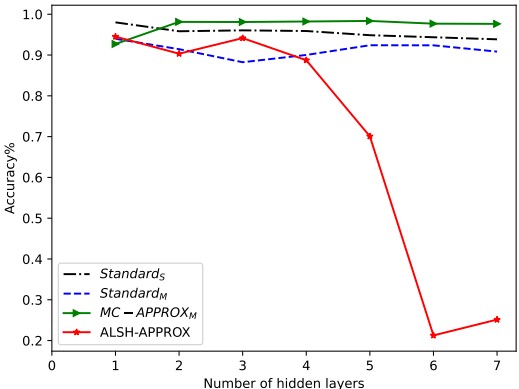 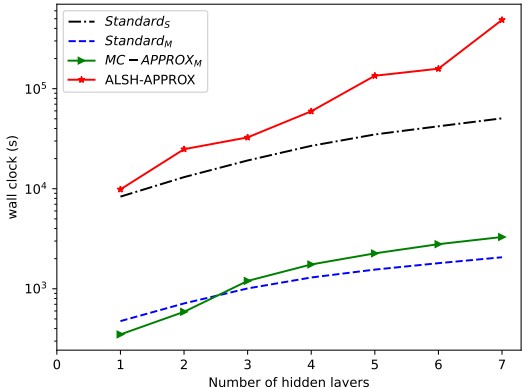

Figure 9: Test accuracy for different numbers of layers.

Figure 10: Total training time for different numbers of layers.

add layers to the network and an increase in the ability to learn nonlinear functions. Clearly, this does not include the cases in which STANDARD overfits.

**ALSH-approx.** The experiment results on ALSH-APPROX (Row 4 in Figure 4) confirm a decrease in accuracy as the number of layers increases. In particular, Figures 4o and 4p show a sharp decrease in performance on 5 to 7 layers. This is confirmed in Figure 9, where the accuracy of ALSH-APPROX drops from 70.07% to 25.14% from 5 to 7 layers. Comparing ALSH-APPROX with STANDARD$_S$ in Figure 4, even though initially the two algorithms performed similarly on a small number of layers (first and second columns), the performance gap quickly increases with the number of layers — confirming the *lack of scalability* of ALSH-APPROX for DNNs. This is also observed in Figure 9.

**MC-approx.** MC-APPROX is designed for use with mini-batch SGD. As we shall further investigate in §10.3, even though MC-APPROX$_S$ outperforms other methods evaluated (Table 3), the runtime for large numbers of layers and datasets is so high that it is infeasible for computation-limited systems. This is reflected in Figure 11. Therefore, as indicated in §9.4, we use mini-batch SGD (with batch size 20) as the default setting in our experiments. The experiment results on MC-APPROX$_M$ are provided in the last row of Figure 4. MC-APPROX$_M$ shows equally good performance across different numbers of layers, confirming its scalability for DNNs. In particular, when varying the number of hidden layers (Figure 9), the minimum accuracy obtained by MC-APPROX$_M$ is 92.71% for one hidden layer. Comparing the confusion matrices of MC-APPROX with ADAPTIVE-DROPOUT and STANDARD in Figure 4, we can see that performance is consistent across the three algorithms. As shown in Figure 9, in most cases, MC-APPROX$_M$ outperformed STANDARD$_M$ with 2% to 4% difference in accuracy. This is also evident in Table 3, where MC-APPROX$_M$ and MC-APPROX$_S$ outperformed other algorithms on the MNIST and Fashion-MNIST datasets with 3 hidden layers. Finally, looking at Figure 9, the only case in which MC-APPROX$_M$ fails to obtain the highest accuracy compared with ALSH-APPROX is when the model has only 1 hidden layer. ALSH-APPROX performs (94.4%) slightly better than MC-APPROX$_M$ (92.71%).

## 10.2 Scalability Evaluation: Time

After studying the impact of network depth on accuracy, we next turn our attention to efficiency (i.e., training time). The results from all five methods on three hidden layers, on one CPU and without parallelization, are summarized in Table 4 and Table 5. Furthermore, Figure 11 provides the runtime during training up to each epoch. Even though from Table 5 it is evident that MC-APPROX$_M$ significantly outperforms other approaches with batch size 20, MC-APPROX$_S$ is slower than ADAPTIVE-DROPOUT$_S$, STANDARD$_S$, and

DROPOUT$_S$. Similarly, Figure 10 shows the runtime comparison of MC-APPROX$_M$ and ALSH-APPROX with STANDARD$_S$ and STANDARD$_M$ (baseline) for different numbers of layers. The results confirm the superiority of MC-APPROX$_M$ over the other algorithms up to 3 layers. Note that the observed increase in the training time of ADAPTIVE-DROPOUT per epoch in comparison to STANDARD can be attributed to the additional computational overhead of the construction of dropout masks and their subsequent multiplication with the weight matrices in each layer (Table 5).

Table 4: Training time per epoch (sec.) with 3 hidden layers and batch size 1 on MNIST.

| Method | ALSH-APPROX | MC-APPROX$_S$ | DROPOUT$_S$ | ADAPTIVE-DROPOUT$_S$ | STANDARD$_S$ |
|---|---|---|---|---|---|
| Total | $807.50 \pm 22.92$ | $422.23 \pm 3.36$ | $196.15 \pm 0.55$ | $225.85 \pm 1.91$ | $361.51 \pm 5.13$ |
| Feedforward | $168.02 \pm 3.34$ | $28.44 \pm 0.076$ | $32.32 \pm 0.04$ | $59.03 \pm 0.10$ | $28.976 \pm 0.04$ |
| Backpropagation | $356.16 \pm 7.28$ | $110.98 \pm 0.69$ | $131.02 \pm 0.57$ | $132.68 \pm 1.89$ | $61.88 \pm 1.02$ |

Table 5: Training time per epoch (sec.) with 3 hidden layers and batch size 20 on MNIST.

| Method | MC-APPROX$_M$ | DROPOUT$_M$ | ADAPTIVE-DROPOUT$_M$ | STANDARD$_M$ |
|---|---|---|---|---|
| Total | $25.426 \pm 0.636$ | $18.458 \pm 0.023$ | $22.748 \pm 0.274$ | $20.16 \pm 0.12$ |
| Feedforward | $2.219 \pm 0.010$ | $2.910 \pm 0.000$ | $6.645 \pm 0.004$ | $2.202 \pm 0.004$ |
| Backpropagation | $9.585 \pm 0.055$ | $7.036 \pm 0.018$ | $7.268 \pm 0.275$ | $3.644 \pm 0.050$ |

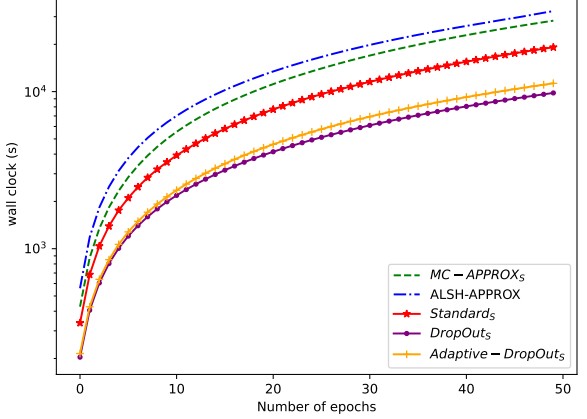

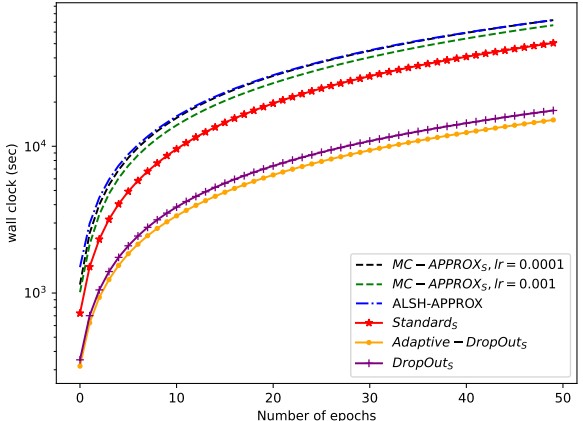

Figure 11: Cumulative training time for each method with 3 hidden layers.

Figure 12: Cumulative training time for each method with 7 hidden layers.

**ALSH-approx.** Before discussing our efficiency results from ALSH-APPROX, let us emphasize that ALSH-APPROX is a scalable algorithm that significantly *benefits from parallelization*. During training, the hash table construction, computing hash signature, querying hash tables, and updating weight vectors by sparse weight gradients are parallelized, which makes the algorithm fast with parallel processing using multiple processing units. We refer interested readers to the details and results of Spring & Shrivastava (2017).

ALSH-APPROX needs to reconstruct the hash tables after a set of weight updates. Following the original implementation of ALSH-APPROX, in our experiments, for the first 10000 training data points, we reconstruct hash tables every 100 images. Then gradually, we expand the set to avoid time-consuming table reconstructions and update the tables every 1000 images. This helps with directing the gradient and decreasing the redundancy in the dataset.

Looking at Table 4 and Figure 11, compared with DROPOUT$_S$, ALSH-APPROX performs better, but it does not outperform STANDARD$_S$ and ADAPTIVE-DROPOUT$_S$. Also, in models with additional hidden layers, we

can see an increase in training time as shown in Figure 10 that is larger than other methods on the same network structure. This is consistent with the results presented by Spring & Shrivastava (2017), where it is shown the runtime significantly drops with parallelization. Evidently, as shown by Spring & Shrivastava (2017), parallelization has no impact on the accuracy of ALSH-APPROX. In other words, the accuracy scalability results of ALSH-APPROX discussed in the previous section are independent of parallelization.

**MC-approx.** Due to the sampling ratio of MC-APPROX ($p \approx 0.1$), MC-APPROX performs more atomic scalar operations than ALSH-APPROX with 5% of the nodes. However, based on the results from Figure 10 and Table 4, MC-APPROX$_M$ and MC-APPROX$_S$ are around 20 and 2 times faster than ALSH-APPROX, respectively. This is because of the significantly lower overhead of MC-APPROX compared to ALSH-APPROX. Figure 10 demonstrates that MC-APPROX$_M$ outperforms ALSH-APPROX and STANDARD$_S$ on time, while having comparable training time with STANDARD$_M$. For networks with fewer than 3 layers, MC-APPROX$_M$ is more efficient than STANDARD$_M$, and for deeper networks, the opposite is true. Nevertheless, Figure 9 confirms the higher accuracy of the MC-APPROX$_M$ for various numbers of layers on MNIST. From Figure 11, it is evident that STANDARD$_S$ is faster than MC-APPROX$_S$ across different epochs. The reason is that, in order to estimate probabilities based on Equation 8 for each mini-batch, MC-APPROX makes a pass over the mini-batch and the matrix $W$. As a result, in SGD, where mini-batch size is one, the overhead time and the time to approximate the matrix multiplication exceeds the required time for exact multiplication (STANDARD$_S$). Finally, we evaluate the algorithms on our deepest network with 7 hidden layers (Figure 12). Similar to our experiment on 3 hidden layers (Figure 11), this experiment once again confirms that unlike MC-APPROX$_M$ (Figure 10)MC-APPROX$_S$ does not scale to deep networks.

## 10.3 Hyperparameters

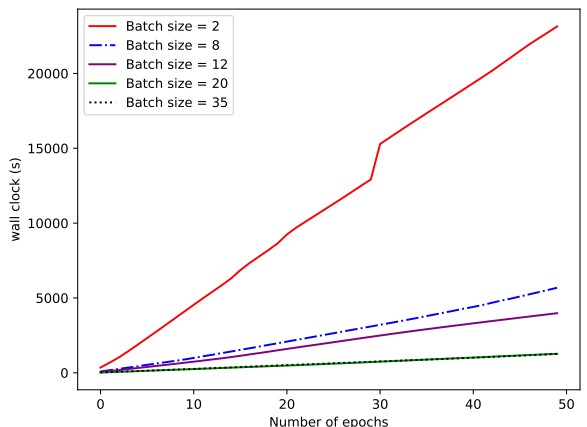 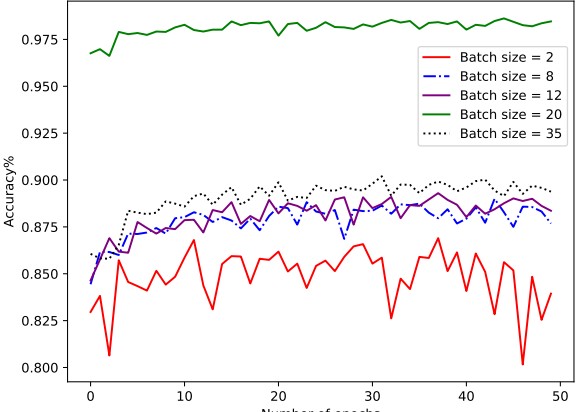

Figure 13: MC-APPROX: Cumulative training time for different batch sizes (learning rate = 0.001).

Figure 14: MC-APPROX: validation accuracy for different batch sizes (learning rate = 0.001).

MC-APPROX is designed for mini-batch stochastic gradient descent, as it uses the set of samples in the mini-batch for error estimation in Equation 8 to identify which rows in $W^k$ to select. In SGD, the estimations would be made using only one sample, and hence are not reliable. As a result, while MC-APPROX performs well for mini-batch SGD with a large-enough batch size (20), its efficiency drops significantly for SGD. While it is known that smaller batch sizes lead to better generalization, in addition to being slower than STANDARD$_S$ (Figures 11 and 12), MC-APPROX$_S$ is prone to overfitting in the stochastic setting.

To evaluate this, we run experiments on stochastic setting where batch size is set to 1 (MC-APPROX$_S$) along with different batch sizes. The results are provided in Figures 7, 8, 13, and 14 and Tables 4 and 5. The results in Figure 14 show the decrease in accuracy for small mini-batches with the same learning rate: the accuracy drops from 98% to 84%. As shown by Shallue et al. (2019), the optimal learning rate to use depends on the

batch size and is smaller for smaller batch sizes. Accordingly, to resolve the overfitting in MC-APPROX$_S$, we decreased the learning rate from $10^{-3}$ to $10^{-4}$. As shown in Table 3 and Figure 7, MC-APPROX$_S$ performs well in terms of accuracy. Moreover, as the model gets more complex by adding hidden layers (Figure 8) and the variance increases, the model is unable to avoid overfitting even with decreasing the learning rate. We discussed in §6.2 that MC-APPROX chooses row-column pairs from matrices such that the columns are from the first input matrix $X \in \mathbb{R}^{m \times n}$ and the corresponding rows are sampled from the second matrix $W \in \mathbb{R}^{n \times n}$. Figure 8 provides evidence of the lack of scalability of MC-APPROX$_S$ in the context of deep networks. This can be attributed to the specific sampling procedure employed by MC-APPROX$_S$. In the stochastic setting, $X_{:,j}$ is reduced to a singleton set. As a result, the time overhead increases, while the reliability of probability estimation for row-column selection decreases.

## 11 Lessons and Discussion

### 11.1 Feedforward Approximation Scalability

A major takeaway in this paper is the negative impact of approximation during the feedforward process. First, in §7, we theoretically analyzed the error propagation effect from layer to layer. In particular, Theorem 2 highlights the *exponential* increase of gradient estimation error in ALSH-APPROX as the number of hidden layers increases. As a result, for neural networks with more than 3 hidden layers, the error can become even larger than the estimation value. Consequently, the gradient estimation can become utterly arbitrary, resulting in completely inaccurate weight updates during the backpropagation step, which leads to an inaccurate model. For MC-APPROX, Adelman et al. (2021) did not observe consistent behaviour across different models in their experiments. Interestingly, the authors provide the theoretical result that (i) approximating both feedforward and backpropagation operations leads to biased estimates, and (ii) approximating only feedforward operations is unbiased. However, their method for feedforward approximation *failed* in experiments (Adelman et al., 2021). As a result, MC-APPROX (the algorithm evaluated in this paper) only adds approximation during backpropagation. We observed similar results for ALSH-APPROX in §10. It is evident in Figures 4, 5, 8, and 9, that ALSH-APPROX failed to scale for DNNs, confirming our theoretical analysis in Theorem 2.

Fortunately, backpropagation optimization can significantly improve training time (Goodfellow et al., 2016; Sun et al., 2017). In our experiments in Table 4 and 5, we observed that backpropagation generally took significantly longer than the feedforward step. As a result, introducing approximation only during the backpropagation step still has the potential to reduce the training time significantly. Nevertheless, designing scalable sampling-based algorithms that introduce approximation on both feedforward and backpropagation in DNNs on CPU machines remains an open research direction.

### 11.2 DNNs and Small Batch Size

As observed in our theoretical analysis and experiment results, ALSH-APPROX does not scale to DNNs with more than a few hidden layers. MC-APPROX, on the other hand, fully scales for DNNs but it is designed based on mini-batch gradient descent and performs well when the batch size is reasonably large (greater than 10). However, the performance of MC-APPROX quickly drops for small batch sizes under the same setting. Particularly, in our experiments, we observed a swift drop in time efficiency (Figure 13) under SGD (when batch size is 1). While MC-APPROX$_S$ demonstrated a high accuracy in some cases (Table 3), this comes at a cost of a significant increase in training time (even compared to STANDARD$_S$) and a high risk of overfitting, especially for deep networks (Figure 8). In summary, designing scalable sampling-based algorithms for SGD on CPU remains an open research direction.

### 11.3 ALSH-approx Prediction in DNNs

We would like to conclude this section with an interesting observation on ALSH-APPROX. Let us consider the confusion matrices of ALSH-APPROX in Figure 4 (Row 4) once again. From Figure 4m, one can confirm that (i) there is no class imbalance in the test set (approximately same number of samples in each class), and (ii) having high accuracy, the model predictions are uniformly distributed across different classes (approximately same number of samples predicted to be in each class). On the other hand, in Figure 4p, it seems not only the

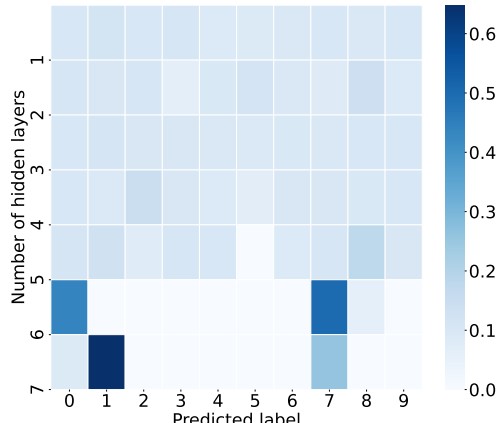

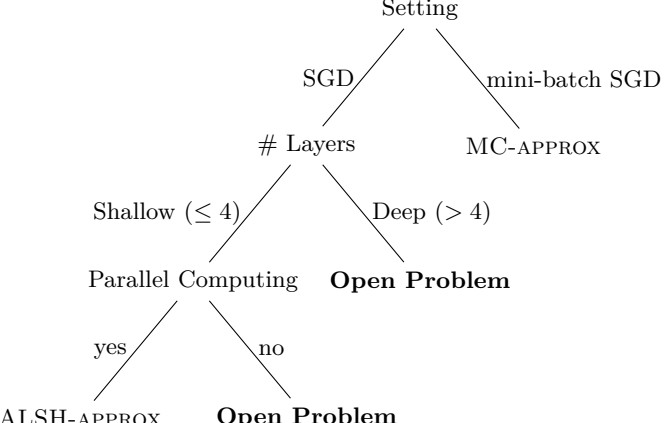

Figure 15: ALSH-APPROX: impact of network depth on label prediction ratio distribution.

Figure 16: Which method to use for training DNNs on CPU.

model is inaccurate, but interestingly only a few labels from the class labels are generated in predictions (all samples are labeled as either 0, 1, or 7). We repeated the experiment multiple times and observed consistent behavior. Furthermore, comparing Row 2 (DROPOUT) with Row 4 (ALSH-APPROX) in Figure 4, we note that, while both methods failed to scale with the number of hidden layers, DROPOUT maintains the label diversity in its prediction, which demonstrates randomness. To better present this, in Figure 15, we provide the ratio of test-set samples predicted for each label (column) for networks with various numbers of hidden layers. It is clear that, while initially the label prediction distribution is uniform, as the number of layers increases, the predictions get concentrated around a few arbitrary classes. The reason is that, while training the model using ALSH-APPROX, as the gradient estimation error increases for deeper networks, a small subset of nodes remains active in deeper layers, regardless of the input sample. As a result, the set of edges for which the weights get updated remains almost the same. Therefore, when predicting the label of an input sample, the same set of nodes is "activated", resulting in a small set of predictions generated.

### 11.4 Optimal Choice of Training Method

Following our findings and lessons learned in this study, in Figure 16 we provide a decision tree to help users decide which method to choose for training DNNs on CPU machines. First, under mini-batch SGD, MC-APPROX has a marginal advantage over the baselines, particularly in terms of accuracy. Under SGD, when the network is not deep (at most four layers) and parallel computing on multiple CPUs is being used, ALSH-APPROX is preferred. On the other hand, in large-scale settings where there are more than four layers or when enough CPU cores for heavy parallelization are not available, the problem of deciding which training method to use is still open.

## 12 Conclusion

This work presents a scalability evaluation of sampling-based approaches for efficient training of deep neural networks with limited computation and memory resources. Alongside our theoretical findings, our experimental results demonstrate a correlation between the number of hidden layers and approximation error in DNNs under hashing-based methods. In addition, we provide valuable insights into the performance of fast training methods in different settings and highlight areas for further research.

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

## A Proofs

*Proof of Lemma 1.* We want to show that the estimation error for the node $n_j^k$ by ALSH-APPROX is as follows.

$$e_j^k = \begin{cases} \sum_{i \notin \uparrow_j^1} x_i W_{i,j}^1 & \text{if } k = 1 \\ e^{k-1} W_{:,j}^k + \sum_{i \notin \uparrow_j^k} \bar{a}_i^{k-1} W_{i,j}^1 & \text{otherwise} \end{cases}$$

First, for $k = 1$:

$$a_j^1 = \sum_{i=1}^n x_i W_{i,j}^1 = \sum_{l \in \uparrow_j^k} x_l W_{l,j}^1 + \sum_{i \notin \uparrow_j^k} x_i W_{i,j}^1 = \bar{a}_j^1 + \sum_{i \notin \uparrow_j^k} x_i W_{i,j}^1$$

$$\implies e_j^1 = a_j^1 - \bar{a}_j^1 = \sum_{i \notin \uparrow_j^k} x_i W_{i,j}^1$$

Analogously, when $k > 1$:

$$a_j^k = \sum_{i=1}^n a_i^{k-1} W_{i,j}^{k-1} = \sum_{i=1}^n (\bar{a}_i^{k-1} + e_i^{k-1}) W_{i,j}^{k-1}$$

$$= \sum_{l \in \uparrow_j^k} \bar{a}_l^{k-1} W_{l,j}^{k-1} + \sum_{i \notin \uparrow_j^k} \bar{a}_i^{k-1} W_{i,j}^{k-1} + \sum_{i=1}^n e_i^{k-1} W_{i,j}^{k-1}$$

$$= \bar{a}_j^k + e^{k-1} W_{:,j}^k + \sum_{i \notin \uparrow_j^k} \bar{a}_i^{k-1} W_{i,j}^{k-1}$$

$$\implies e_j^1 = a_j^1 - \bar{a}_j^1 = e^{k-1} W_{:,j}^k + \sum_{i \notin \uparrow_j^k} \bar{a}_i^{k-1} W_{i,j}^{k-1}$$

$\square$

*Proof of Theorem 2.* For any node $n_p^l$, we have

$$\sum_{i \in \uparrow_p^l} a_i^{l-1} W_{i,p} = c \sum_{i \notin \uparrow_p^l} a_i^{l-1} W_{i,p}.$$

We then use induction to prove $a_j^k = \bar{a}_j^k \left(\frac{c+1}{c}\right)^k$.

*Base case.* When $k = 1$:

$$a_j^1 = \sum_{i=1}^n x_i W_{i,j}^1 = \sum_{l \in \uparrow_j^k} x_l W_{l,j}^1 + \sum_{i \notin \uparrow_j^k} x_i W_{i,j}^1 = \bar{a}_j^1 + \frac{1}{c} \bar{a}_j^1 = \bar{a}_j^1 \frac{c+1}{c}$$

*Inductive step.* Assuming $a_j^{k-1} = \bar{a}_j^{k-1} \left(\frac{c+1}{c}\right)^{k-1}$:

$$a_j^k = \sum_{i=1}^n a_i^{k-1} W_{i,j}^{k-1} = \sum_{i=1}^n (\bar{a}_i^{k-1} + e_i^{k-1}) W_{i,j}^{k-1} \tag{9}$$

$$= \sum_{l \in \uparrow_j^k} \bar{a}_l^{k-1} W_{l,j}^{k-1} + \sum_{i \notin \uparrow_j^k} \bar{a}_i^{k-1} W_{i,j}^{k-1} + \sum_{i=1}^n e_i^{k-1} W_{i,j}^{k-1}$$

$$= \bar{a}_j^k + \frac{1}{c} \bar{a}_j^k + \sum_{i=1}^n e_i^{k-1} W_{i,j}^{k-1} = \frac{c+1}{c} \bar{a}_j^k + \sum_{i=1}^n e_i^{k-1} W_{i,j}^{k-1}$$

Let $A = \sum_{i=1}^{n} e_i^{k-1} W_{i,j}^{k-1}$. We have

$$e_i^{k-1} = a_i^{k-1} - \bar{a}_i^{k-1} = a_i^{k-1} - a_i^{k-1}\left(\frac{c}{c+1}\right)^{k-1}$$
$$= a_i^{k-1}\left(1 - \left(\frac{c}{c+1}\right)^{k-1}\right).$$

Thus,

$$A = \sum_{i=1}^{n} e_i^{k-1} W_{i,j}^{k-1} = \sum_{i=1}^{n} a_i^{k-1}\left(1 - \left(\frac{c}{c+1}\right)^{k-1}\right) W_{i,j}^{k-1}$$
$$= \left(1 - \left(\frac{c}{c+1}\right)^{k-1}\right) \sum_{i=1}^{n} a_i^{k-1} W_{i,j}^{k-1} = \left(1 - \left(\frac{c}{c+1}\right)^{k-1}\right) a_j^{k}.$$

Now, plugging $A$ back into Equation 9, we get:

$$a_j^k = \frac{c+1}{c}\bar{a}_j^k + A = \frac{c+1}{c}\bar{a}_j^k + \left(1 - \left(\frac{c}{c+1}\right)^{k-1}\right) a_j^k$$
$$\implies a_j^k\left(1 - \left(1 - \left(\frac{c}{c+1}\right)^{k-1}\right)\right) = \frac{c+1}{c}\bar{a}_j^k \implies a_j^k = \bar{a}_j^k\left(\frac{c+1}{c}\right)^k$$

$\square$

