# OpenReview forum: "Sampling-Based Techniques for Training Deep Neural Networks with Limited Computational Resources: A Scalability Evaluation"
_TMLR — Rejected by TMLR_

### Review · Reviewer_USen · 2023-09-02

**Summary Of Contributions:**

The paper is about evaluating the scalability of sampling-based techniques for training deep neural networks (DNNs) on CPU machines with limited computational resources. The authors provide a theoretical analysis showing that feedforward approximation is an obstacle against scalability. They conduct comprehensive experimental evaluations that demonstrate the most pressing challenges and limitations associated with the studied approaches. They observe that the hashing-based node selection method is not scalable to a large number of layers, confirming their theoretical analysis. Finally, they identify directions for future research.

**Audience:**

Yes

**Broader Impact Concerns:**

No potential negative social impact concerns.

**Claims And Evidence:**

No

**Requested Changes:**

I think the authors need to carefully discuss the practical benefits of this work. As I mentioned in weakness, focusing on the acceleration of training on such small-scale datasets & models is not intriguing as the computing resources nowadays are more accessible.

**Strengths And Weaknesses:**

Strength:

1. The structure of this paper is well-organized and the paper is easy to read.

2. The authors incorporated several different approaches into a framework to analyze.

Weakness:

1. The scope of this paper, as stated by the authors, is training fully connected layers-based neural networks (mostly) on CPUs. While such scenarios seem to be an important field as the resource is really limited. It is really hard to see optimizing training on such small-scale networks and datasets can have an impact in the real world. For example, to train a neural network on MNIST, and Fashion-MNIST, we can use the free resources in Google Colab where a free GPU can be provided to accelerate the training.

2. The authors list several related works, however, most of them are from the middle 2010s. I was wondering whether this particular field of research has been important in recent years.

3. While I understand the training scenarios are different here, I might have several different opinions with the authors:
- "The computation bottleneck in the training of a DNN is matrix multiplication." The matrix multiplication is highly optimized in today's computing hardware. However, the IO bottleneck (data loading, write-and-read) seems to be more severe and restricts the training speed. The datasets used in this paper rarely have this problem. But as I mentioned in weakness 1, this type of task is more like a toy problem.

- "In many cases, Dropout improved the runtime efficiency compared to the standard training process on the same architecture." Dropout introduces an irregular pattern in sparsity. This cannot accelerate the training speed, at least in the parallel computing (e.g., SIMD) manner.

4. The sampling-based method proposed in this method seems like cannot generalize to large-sale neural networks on a more complex dataset. Since the accuracy of CIFAR-10 is 62.1.

---

> ### Author Response · Authors · 2023-09-25
>
> 1. We appreciate the provided feedback. Cloud-based computing services offer constrained resources in their free tier, encompassing CPU, TPU, and T4 GPU allocations. Even in the basic premium plan, access to GPUs suitable for computationally intensive tasks such as training LLMs is limited, with only the V100 GPU available. Additionally, Google Colab Pro's plan grants users 100 compute units per month, with a supplementary charge of $10 per 100 compute units exceeding this limit. Access to more robust GPUs, such as A100, incurs even higher expenses. In situations involving the training of substantial models or iterative experiments, this approach is cost-prohibitive for many users. Consequently, when assessing existing methodologies, we were compelled to select benchmark datasets and configurations that remained within the constraints of an average user’s available resources.
>
> 2. Regarding the gap in related work, the domain of training DNNs on CPU-based machines remains relatively unexplored within the research landscape. This niche presents a promising avenue for future work. We aim to draw attention to this domain, shedding light on the associated challenges and prospective research directions.
>
> 3. Our paper places particular emphasis on two studies published in 2021 and 2017, aiming to reference works that have had a significant influence on this domain. We have also cited more recent papers in our manuscript. In the revised submission, we have added an additional set of citations of recent papers.
>
> 4.We acknowledge that the deployment of powerful GPUs in cloud-based environments, optimized for vectorized operations, presents a formidable advantage for training deep neural networks on an expansive scale. In these contexts, the management of input/output (IO) operations and network communication can indeed introduce novel challenges. However, it is important to note that, at least in the case of CPU machines and their present operational constraints, computational efficiency remains the primary bottleneck [2]. We share the vision of developing efficient algorithms to overcome these computational challenges on CPU machines. Should these algorithms succeed in scaling to very-large settings, IO and network communication may become the most prominent obstacles.
> Within the domain of DL, particularly in the analysis of conv. 2D images and videos, the computation of features through matrix mul. persists as the predominant bottleneck [3]. Under specific scenarios, users may encounter limitations pertaining to memory capacity while necessitating the execution of matrix muls involving a substantial sparse matrix and an equivalently voluminous dense matrix. Nevertheless, techniques such as MC-Approx have been devised to mitigate data or parameter size, alleviating the bottleneck associated with CPU-to-GPU data transfers.
> Both directions, whether focused on GPU-centric or CPU-centric methodologies, share a common focal point in tackling challenges linked to matrix multiplication [4]. Furthermore, in certain instances, optimization efforts in arithmetic operations can contribute to the implementation of tiling strategies and the effective harnessing of available hardware resources.
>
> [2] Y Ziyu, et al. "EdgePC: Efficient Deep Learning Analytics for Point Clouds on Edge Devices." ISCA. 2023
> [3] T Liang, et al. “Pruning and quantization for deep neural network acceleration: A survey”, Neurocomputing, 2021
> [4] S Dalton, et al. Optimizing Sparse Matrix—Matrix Multiplication for the GPU. ACM Trans. Math. Softw., 2015
>
> 5. ALSH-Approx generates a sparsely active set of neurons in a random manner, mitigating the likelihood of conflicting gradient updates. Sparse updates are particularly well-suited for asynchronous and parallel gradient updating schemes. However, this does not apply to dropout and adaptive dropout techniques. The sparse activations induced by these algorithms predominantly serve the purpose of enhancing generalization rather than optimizing computational efficiency [5]. It is essential to clarify that the experiment results we are discussing in this paragraph were not executed in a parallel computing setting.
>
> [5]. J Dean, et al. Large scale distributed deep networks. Adv. Neural Inf. Process, 2012.
>
> 6. The results in our study pertaining to the CIFAR-10 dataset diverge from those reported in the original paper due to a distinct experimental configuration. In our investigation, we applied MC-Approx to classifier layers subsequent to ResNet blocks, in contrast to the original paper's focus solely on convolutional layers. This discrepancy in our approach accounts for the disparity in results and the consequential reduction in performance relative to the original findings.
> The pursuit of advancing DNN training on CPU machines unveils a compelling avenue replete with practical advantages, which were not emphasized enough in our initial submission. In Section 2 of the revised version, we discuss these benefits.

---

### Review · Reviewer_EkQJ · 2023-09-03

**Summary Of Contributions:**

This paper studies the scalability of activation subsampling methods for efficient training of neural networks through the lens of the connection between gradient approximation error and the number of layers. The paper unifies two main lines of the subsampling method (masked a portion of nodes v.s. replace them with previous layers) into matrix multiplication approximation, and derives a theoretical result showing that the gradient estimation error increases exponentially w.r.t. the number of layers. This indicates that current subsampling-based methods are fundamentally not scalable to deeper neural networks. The theoretical results are further verified by extensive empirical studies on small-scale MLPs and ConvNets across various toy datasets (MNIST-variants, NORB, and CIFAR-10).

**Audience:**

Yes

**Claims And Evidence:**

Yes

**Requested Changes:**

I would encourage the authors to consider empirical evaluation on large-scale networks and datasets, but this is only a suggestion, not a requirement.

**Strengths And Weaknesses:**

[Strength]

1. [major] The paper provides a novel theoretical view connecting the approximation error of subsampling-based methods with the numbers of the layers, showing that the former increases exponentially w.r.t. the latter. These results demonstrate that the existing subsampling-based method fundamentally does not scale to deeper networks
2. [major] The paper provides an extensive empirical evaluation to support its theory. The experiments take into account multiple sampling-based methods, datasets, architectures, and Hyperparameters for the optimization process.
3. The paper is well-organized; The presentation is clear and easy to follow.

[Weakness]

1. [minor] The empirical evaluation focuses mostly on toy networks and datasets. While I acknowledge the fact that the paper focuses on resource-limited settings, I think extending the empirical validation to moderate to large-scale scenarios is technically possible and would further strengthen the paper.

---

> ### Author Response · Authors · 2023-09-25
>
> We would like to thank the reviewer for their diligent review of our paper and for their valuable recommendation. In response to their insightful comment, we conducted supplementary experiments involving two distinct datasets. Firstly, we explored the USPS dataset, thereby introducing a variation from the commonly employed MNIST dataset. Furthermore, as per the reviewer's suggestion, we extended our experimentation to include a substantially larger dataset consisting of 1 million MNIST samples.
>
> Due to the constraints of the time frame available, only one of these experiments, specifically involving MC-Approx with a mini-batch size of 20, successfully concluded, achieving an impressive accuracy rate of 99.99%. It is pertinent to note that the outcomes from the stochastic MC-Approx and ALSH-Approx experiments were obtained after 19 and 1 iteration(s), respectively. Specifically, Stochastic MC-Approx yielded a 99.17% accuracy rate after 1.96 * 10^5 seconds, while ALSH-Approx delivered an 87.73% accuracy rate after a single iteration spanning 1.6 * 10^4 seconds.
>
> In the event of an opportunity for revision, we eagerly anticipate supplementing the paper with comprehensive results from the remaining experiments as they reach completion.

---

### Review · Reviewer_poLY · 2023-09-11

**Summary Of Contributions:**

The paper mainly discusses the negative impact of approximation in both the feedforward and backpropagation process in deep neural networks. There are two main contributions including theoretical analysis and comparison of existing approximation methods. The authors provide a theoretical analysis of error propagation in approximation methods, particularly focusing on the exponential increase in gradient estimation error with the number of hidden layers. They emphasize that as the number of hidden layers exceeds three, the error can surpass the estimation value, leading to highly inaccurate weight updates during backpropagation and ultimately resulting in inaccurate models. The author also discusses the comparison of two approximation methods, MC-approx and ALSH-approx. This paper highlights the importance of understanding the behavior of approximation methods in deep neural networks to mitigate their negative impact on model accuracy.

**Audience:**

Yes

**Broader Impact Concerns:**

I have no concerns on the ethical implications of this work.

**Claims And Evidence:**

Yes

**Requested Changes:**

Regarding the CPU and GPU discussion, what are the differences or distinctions in how these methods perform on CPU versus GPU? Most of the datasets related to MNIST, which this paper focuses on, can typically be trained directly on a CPU. Therefore, a more specific analysis is needed regarding the discussed methods and the specific implementation challenges on CPU and GPU. Currently, it appears that these methods and datasets can be effortlessly executed on both CPU and GPU platforms.

The legend in Figures 5 and 6 should be refined for better clarity and understanding.

Addressing the questions in the "weaknesses" part could provide readers with a clearer understanding of the limitations and potential areas for improvement in the research.

**Strengths And Weaknesses:**

Strengths:

The motivation of this paper is clear, and the article's structure is well-organized, allowing readers to easily follow the content from the introduction of several sampling-based techniques to experimental validation.

Weaknesses:

The authors consistently emphasize their goal to assess the scalability of these approaches on CPU machines with limited computational resources. However, these methods are applicable and effective on both CPU and GPU platforms.

This paper claims that "We make a connection between two separate sampling-based research directions for training DNNs by showing that both techniques can be viewed as special cases of matrix approximation, where one samples rows of the weight matrix while the other sample its columns. To the best of our knowledge, there is no previous work in the literature to make this observation". I think the relationship between such techniques and matrix approximation is quite common in network pruning tasks [1].

[1] Ma, Yuzhe, et al. "A unified approximation framework for compressing and accelerating deep neural networks." 2019 IEEE 31st International Conference on Tools with Artificial Intelligence (ICTAI). IEEE, 2019.

The Section 5.2 claims that "Therefore, Adelman et al. (2021) propose a new sampling distribution that yields an unbiased estimate of the weight gradient ∇wL when it is used only during the feedforward step." Is this estimation approximation used in feedforward step or backpropagation (as shown in Section 6 "Adelman et al. (2021) already observed the low performance of MC-approx when the feedforward step is approximated and therefore only applied approximation during backpropagation for MLPs.")?

One small question about the proof of Lemma 1, does ALSH-approx need a coefficient to approximate the numerical value of the inner product of all <a, W>, just like the coefficient 1/cpi in Monte Carlo method? Or the ALSH-approx merely selects the small part of w∗ ∈ W, and use <a, w∗> to approximate <a, W>? As shown in Section 6, "suppose c = 5, i.e., the weighted sum for the active nodes is five times that of the inactive nodes", what if we use the coefficient to better compensate for the missing value of inactive nodes? Can this approach alleviate error accumulation?

In Section 7, "In particular, MC-approx incorporates approximation exclusively within the convolutional layers, while maintaining exact computations in the classifier layers." Classifier layers indeed are fully connected layers. Why does MC-approximation lead to precise calculations?

---

> ### Author Response · Authors · 2023-09-25
>
> 1. We appreciate the reviewer's insight. None of these methods are suitable for large-scale models on standard CPUs without parallelization or distributed computing. Spring and Shrivastava (2017) used a machine with 56 cores and 256 GB of memory, while Adelman et al. (2021) employed an Intel Xeon Silver 4210 CPU with four Nvidia V100 GPUs with 32GB of memory. Our study focuses on the perspective of an average user without access to such resources.
>
> 2. We acknowledge the existing body of literature on the utilization of approximation algorithms for the purposes of pruning and expediting neural networks. Our objective in this study is to establish a connection between two prominent methodologies that have been applied to tackle computational challenges within this domain. Specifically, we aim to bridge the gap between two prevalent approaches: (1) techniques that entail the removal of nodes within each layer while computing the precise values for the retained nodes within the same layer, and (2) methods that prioritize the approximation of computations for all nodes within a given layer.
>
> 3. Adelman et al. (2021) showed that while feedforward phase approximation is theoretically unbiased, its effectiveness depends on the neural network and varies with datasets. For example, MC-approx works best during backpropagation with exact feedforward computations for a multi-layer perceptron on MNIST. Conversely, for a WRN-28-10 network on CIFAR-10, optimal performance requires approximation in both phases but with reduced sampling. This gap between theory and experiments is a notable limitation of Adelman et al.'s (2021) approach.
>
> 4. Thank you for reviewing Lemma 1. ALSH-approx selects a subset of nodes as "active" and computes inner products only for these nodes. We've set K and L parameters to 5 and 6, resulting in about 5% active nodes. Adjusting K and L can change this ratio, affecting error ('c'). More active nodes reduce error but increase inner product computations. We are committed to incorporating these findings into the paper should an opportunity for revision become available.
>
> 5. Adelman et al. (2021) restricted the use of approximation exclusively to the convolutional layers while preserving precise computations in the classifier layers. This approach proves particularly advantageous in the context of GPU-based settings and tensor operations, which are the central focus of their study. By applying approximation selectively to the convolutional layers, they achieve the desirable outcome of consolidating the typically sparse convolution matrices, thereby facilitating computationally efficient GPU operations for these convolutional layers. Our experimental observations revealed a noticeable degradation in performance when the MC-Approx method is applied to the fully connected layers within CNNs. In particular, the performance on the CIFAR-10 dataset is reduced significantly.
>
> 6. We are committed to diligently addressing all requested modifications. In the context of the CPU and GPU discussion, it is essential to clarify that LSH-Approx has been purposefully crafted for CPU utilization and is not intended for GPU-based operations. Its functionality hinges on the creation of Locality-Sensitive Hashing (LSH) buckets and subsequent LSH lookup to identify active nodes. This approach leverages parallelization across a multitude of CPU cores, which constitutes the sole hardware setting employed in the original paper's experimental validation. Conversely, MC-Approx, as indicated by its paper title, is primarily oriented towards GPU tensors. Nonetheless, it exhibits versatility in its adaptability to CPU environments. Both LSH-Approx and MC-Approx adopt a sampling-based strategy to selectively perform a fraction of the computationally intensive matrix multiplications. It is noteworthy that this sampling methodology is executed in two distinct and orthogonal manners, rendering them apt choices for inclusion in our evaluation framework.
>
> 7. We have incorporated experimental findings from an additional dataset (the USPS dataset) into our study. This study primarily centers its attention on the architectural aspects of neural networks, with a specific emphasis on their depth. However, as detailed in Section 7, it is crucial to acknowledge that Locality-Sensitive Hashing Approximations (LSH-Approx) encounter inherent limitations when applied to convolutional layers. This limitation imposes restrictions on our capacity to effectively leverage expansive datasets such as ImageNet.The latter is typically employed for assessing the performance of algorithms in the context of convolutional layers and distributed computing scenarios, as acknowledged in previous research (Adelman et al. (2021)). Consequently, we confined our experiments to configurations compatible with the computational capabilities of all the assessed algorithms, ensuring a fair evaluation of their scalability.
>
> 8. Figures 5 and 6 have been updated. Thanks.

---

### Decision · Action_Editor_b1Fi · 2023-10-23

**Recommendation:** Reject

**Comment:**

Reviewers' comments highlight important concerns regarding the relevance and appeal of the paper's findings to the audience of TMLR. Reviewers point out that the paper's experimental design, which includes the use of MNIST/CIFAR-10 datasets and shallow neural networks, may not be suitable for evaluating fast training methods in what are commonly considered "large-scale settings." This raises questions about the practical applicability of the paper's findings to real-world scenarios. The authors should consider expanding their experiments to include more extensive evaluations, potentially with larger datasets and more complex neural network architectures, to enhance the practical value and generalizability of their findings.

**Audience:**

It appears that the findings presented in this paper may have limited appeal to TMLR's audience. The reviewers' comments highlight concerns about the compatibility of the paper's experimental setup with "large-scale settings." They question the choice of datasets (MNIST/CIFAR-10) and neural networks of fewer layers for evaluating fast training methods. Additionally, the reviewer expresses concerns about the practicality and relevance of the paper's proposed method, especially when compared to free cloud-based GPU training services. These concerns suggest that the findings may not be of interest to individuals seeking solutions for most real-world, moderate-sized AI model training, though authors also realized that there is a fast-growing interest in utilizing deep neural networks in large-scale settings.

**Claims And Evidence:**

The claims made in the submission, regarding speeding up the training time of deep neural networks by approximating matrix products, are not sufficiently supported by accurate, convincing, and clear evidence. The reviewers express concerns regarding the suitability of the datasets (MNIST/CIFAR-10) and neural networks of three layers for representing "large-scale settings." The reviewers believe these settings do not align with the concept of large-scale training, which raises questions about the relevance of the claims. The authors reference "Faster Neural Network Training with Approximate Tensor Operations" (Adelman et al. (2021)) as relevant work. However, it is instructive to note that more extensive experiments have been conducted to evaluate the sampling algorithm in (Adelman et al. (2021)), such as using Wide ResNet 28-10 on CIFAR-10 and ResNet-50 and ResNet-152 on ImageNet. In contrast, the MNIST-level-sized dataset and networks with only three layers in the authors' experiments are insufficient to adequately evaluate the effectiveness of the algorithm.